# Fairy circles in Namibia are assembled from genetically distinct grasses

Christian Kappel [1], Nicola Illing [2], Cuong Nguyen Huu[1], Nichole N. Barger [3], Michael D. Cramer [4], Michael Lenhard [1✉] & Jeremy J. Midgley [4✉]

Fairy circles are striking regularly sized and spaced, bare circles surrounded by *Stipagrostis* grasses that occur over thousands of square kilometres in Namibia. The mechanisms explaining their origin, shape, persistence and regularity remain controversial. One hypothesis for the formation of vegetation rings is based on the centrifugal expansion of a single individual grass plant, via clonal growth and die-back in the centre. Clonality could explain FC origin, shape and long-term persistence as well as their regularity, if one clone competes with adjacent clones. Here, we show that for virtually all tested fairy circles the periphery is not exclusively made up of genetically identical grasses, but these peripheral grasses belong to more than one unrelated genet. These results do not support a clonal explanation for fairy circles. Lack of clonality implies that a biological reason for their origin, shape and regularity must emerge from competition between near neighbor individuals within each fairy circle. Such lack of clonality also suggests a mismatch between longevity of fairy circles versus their constituent plants. Furthermore, our findings of lack of clonality have implications for some models of spatial patterning of fairy circles that are based on self-organization.

[1] Institute for Biochemistry and Biology, University of Potsdam, Karl-Liebknecht-Str. 24-25, D-14476 Potsdam-Golm, Germany. [2] Department of Molecular and Cell Biology, University of Cape Town, Rondebosch 7701, South Africa. [3] Department of Ecology and Evolutionary Biology, University of Colorado at Boulder, 1900 Pleasant St, Boulder, Colorado, CO 80302, USA. [4] Department of Biological Sciences, University of Cape Town, Rondebosch 7701, South Africa. ✉email: michael.lenhard@uni-potsdam.de; jeremy.midgley@uct.ac.za

Fairy circles (FCs) are remarkably circular and regular vegetation patterns covering an area of thousands of square kilometres in hyper-arid grasslands in Southern Africa (e.g. Namibia)[1–3]. They are typically made up of perennial tufts of *Stipagrostis* grass growing around barren circular patches roughly 2–10 m in diameter, with annual or perennial grasses in the area between circles, the matrix[1,2]. The edges of FCs are about 5–10 m apart and when dense they have a regular hexagonal arrangement[3]. Similar vegetation rings also occur in arid lands in Australia, Israel and elsewhere[1–7]. The processes leading to the formation of FCs have been the subject of debate for several decades[8–13].

Aspects of FCs that require explanation include how and why individual circles form, how they persist and how the similar size and regular pattern is established at the landscape scale. The bare centres of individual FCs persist in situ almost permanently, although the peripheral plants delimiting the circles are much shorter-lived. Van Rooyen et al.[8] noted no change in the position of five marked FCs over 22 years, despite intervening droughts[8]. Tschinkel[14] analysed pairs of aerial photographs taken 4 years apart, and on the basis of how many FC positions were unchanged, died or emerged, estimated that average sized FCs persist in situ for an average of 75 years, implying some circles remain in situ for a century[14]. Similarly, using aerial photographs, Juergens[2] noted a high survival (97%) of FCs in situ over a 50-year period (=0.06% pa mortality), implying an age of millennia for some FCs[2].

Many alternative hypotheses have been proposed for FC spatial and temporal patterns, but without agreement. The first hypothesis is based on ecosystem engineering by termites that remove plants from the centres of circles[2], facilitating localized underground water accumulation in circle centres. This moisture maintains the termites and the band of perennial grass on which termites feed year-round. The spatial patterning of the FCs is considered to result from competition between termite colonies[2]. However, the poor correlation of FCs with the presence of specific termites is an important concern with this hypothesis[13].

The second hypothesis is based on the clonal mode of growth of individuals of many arid-land species that create vegetation rings[4,6,7,15]. For example, rings are formed by one of the Namibian FC species, *Stipagrostis ciliata* in the Negev desert. Here individual plants of this species send out underground rhizomes, which, with increasing age, results in a ring of ramets (i.e. sprouts from the same clonal colony or genet) around a barren central patch, which forms as the plant centre dies[4]. Such vegetation rings form and enlarge centrifugally due to competition between ramets, and as this process continues over time new ramets establish successfully only towards the periphery[4,7]. Globally, all of the many plant species that form circles do so by this type of clonal growth[15]. As the pattern of FCs is virtually fixed in situ for centuries, this suggests that the plants that indicate the pattern may also be long-lived. Individual clonal plants can be extremely long-lived[15,16], which could then match longevities between the plants and the FCs they delimit. The clonality hypothesis could thus explain how circular shapes form (by self-thinning of ramets spreading from source plant), why bare centres occur (resource depletion and source plant death) and how they can persist over long periods of time (by continuous production of short-lived ramets). However, clonality has been disputed[8] as an explanation for FCs for two reasons. Firstly, van Rooyen et al. suggested that the FCs referred to by Danin and Orshan are considerably smaller (about 2 m) than most FCs in Namibia[4,8]. Although this is correct that FCs tend to be much larger in Namibia, the mean size of FCs can be as small as 2.5 m in some areas[2]. Secondly, van Rooyen and colleagues suggested that clonality cannot explain why FC centres are bare[8]. Bare centres are now well known in rings and

are commonly explained as being due to inter-ramet competition and resource depletion[7].

The third hypothesis for FC spatial patterns is that it emerges through vegetation self-organization (the VSO hypothesis)[1,5,12,17,18]. Grasses in the peripheral band outcompete grasses in the FC centre and keep it bare and moist at depth[1]. The plants in the peripheral bands also compete with the matrix grasses and with the peripheral bands on adjacent FCs to maintain the regular pattern[1]. Mathematical vegetation models based on partial differential equations represent the theoretical basis of the VSO hypothesis[17,18]. These partial differential equations do not operate at the scale of individual-plant interactions, but at the level of local processes (largely rates of lateral water movement due to plant evapo-transpiration) in comparison to rates of lateral biomass spread. For these current VSO models, lateral water flow needs to be about 100 times faster than lateral biomass spread[18]. Thus, the mode and rate of how biomass spreads, whether via clonal growth or seed dispersal and population growth[18], has an impact on the viability of vegetation-patterning models. In the absence of such information for FCs, the most recent VSO modelling study[18] assumed a rate of biomass dispersal (1.2 m² yr⁻¹) derived from the Canadian woodlands[19], for clonal spread or seed dispersal. These literature values may be unrealistic for FCs. For example, 1.2 m² yr⁻¹ is likely to be too high for clonal growth in the arid circumstances in which FCs occur. *Stipagrostis* individual plants across the landscape are typically only 0.005–0.13 m² in canopy area with a mean area of 0.05 m² (ref. [20]). Similarly, clonal spread in other systems can also be lower than 1.2 m² yr⁻¹ by orders of magnitude (ref. [19]). Alternatively, if *Stipagrostis* plants are not clones, their highly awned seeds will disperse much further than 1.2 m² yr⁻¹. Finally, if the assumed 1.2 m² yr⁻¹ biomass spread[18] is due to the total growth of new recruits per parent individual, then this implies unrealistically high population growth rates (>20 new recruits, each with 0.05 m² in canopy volume[20] required per parent individual). Getzin et al.[3] acknowledge that in the context of their VSO model it needs to be further investigated whether grass tufts experience a central dieback due to self-thinning, i.e. the role of clonality[33]. Thus, clonality may be relevant to some VSO models.

Juergens has argued that there is a spatial mismatch in the root length and inter-FC distances[10] and therefore that FCs cannot directly interact with each other to produce the regular spatial pattern. An extreme example of this mismatch is a study by Ravi et al. reporting that the roots of peripheral plants in FCs had a mean length of 5.9 cm, which is more than two orders of magnitude shorter than inter-circle distances (10–20 m)[12]. These short root lengths would make competition for water and other resources over long distances crucial for explaining FC patterning[1]. Most recently, Ravi et al.[12] rejected the termite hypothesis due to an absence of termites and partially invoked clonal dynamics as well as the VSO to explain their FC patterning[12].

In summary, there are critical issues with all the hypotheses explaining the formation of FCs and the role of clonality appears to be relevant to two of the hypotheses. Here, we use ddRAD-seq as a genetic test of clonality of peripheral grasses. Our analysis indicates that most individual grasses surrounding FCs are genetically distinct and does not support the clonality hypothesis.

## Results

**Peripheral grasses surrounding Fairy Circles have mixed ploidy levels.** Clonality can only be tested by determining whether all the plants surrounding an FC are genetically identical, together representing one genet with multiple ramets. We set out to test this for FCs formed by two species, *S. ciliata* and *S. uniplumis*, in two different regions of the Namib Desert (Fig. 1a, b;

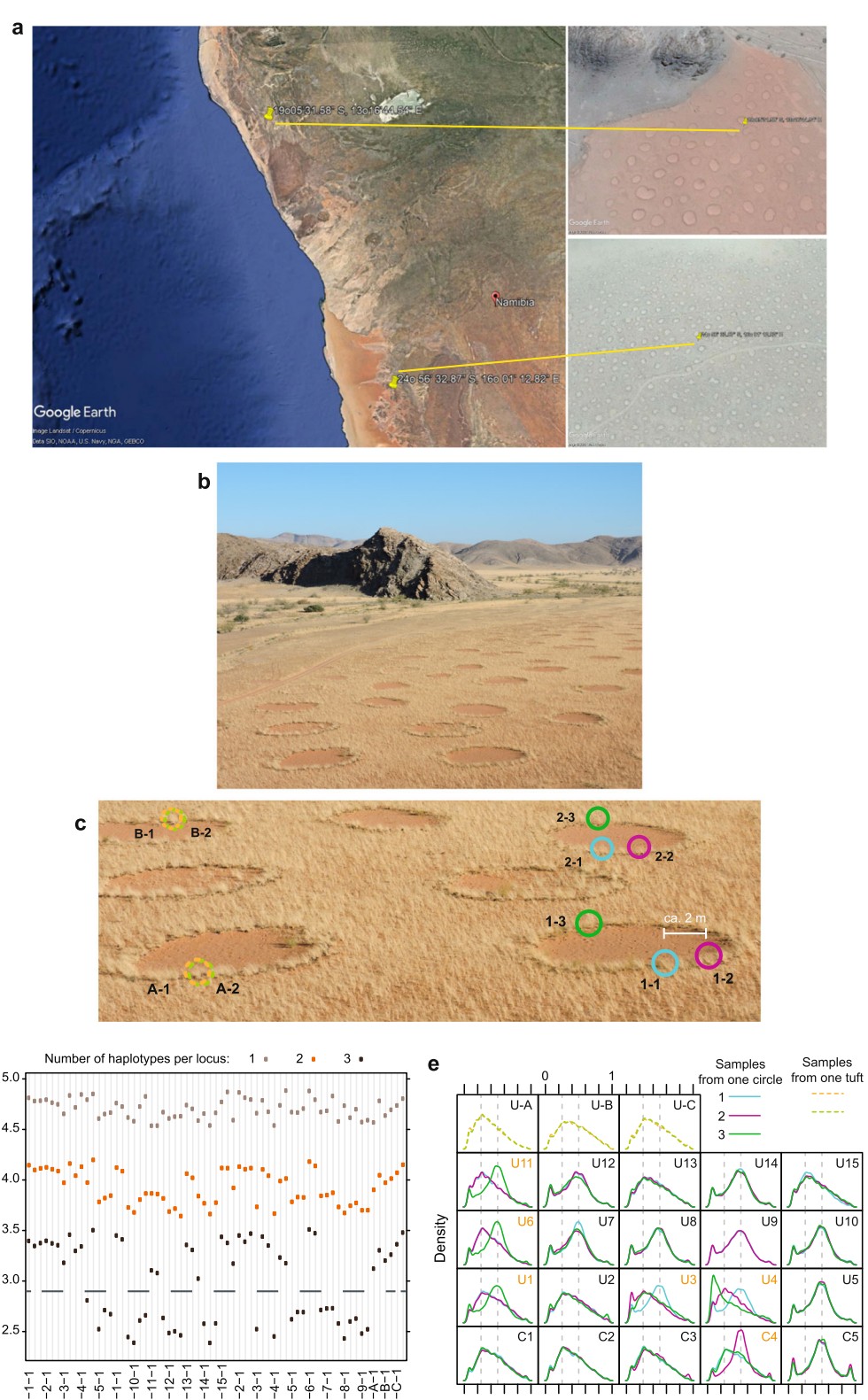

Supplementary Fig. 1). For *S. ciliata* and *S. uniplumis* we sampled five and 15 circles, respectively, collecting leaf material from three individual plants per circle. Two of the plants ('1' and '2') were located within 2 m of each other along the circumference of the peripheral vegetation, while the third ('3') was chosen on the opposite side of the circle (Fig. 1c). For *S. uniplumis*, we took two samples each from one tuft in three unrelated circles ('A' to 'C') to

test for genetic relatedness within a tuft (Fig. 1c). We extracted genomic DNA and performed double-digest restriction-associated DNA sequencing[21], resulting in an average of 63,096 loci ($sd = 15{,}437$) per sample (Fig. 1d). Inspection of the data indicated that there were over 1000 loci with more than two distinct haplotypes for 11 out of the 15 *S. ciliata* samples and 18 out of the 44 *S. uniplumis* samples (Fig. 1d). This number was about

**Fig. 1 The peripheries of Namib Desert fairy circles consist of individuals at different ploidy levels. a** Satellite images annotated from Google Earth of Namibia indicating the two sampling sites. *S. ciliata* was sampled in the more southern, *S. uniplumis* in the more northern site. Fairy circles are visible on the higher-magnification views from Google Earth on the right. **b** Photograph of the study location from which *S. uniplumis* was sampled. **c** Illustration of sampling regime for *S. uniplumis*. Each FC (e.g. circle 1 on bottom right and circle 2 on top right) was sampled three times. Samples '1' (blue circle) and '2' (purple circle) were on the perimeter of the circle, within 2 m of each other (i.e. 1-1 and 1-2 are samples 1 and 2 from circle 1; 2-1 and 2-2 are samples 1 and 2 from circle 2), while sample '3' (green circle) was on the opposite side of the respective circle (1-3 or 2-3). Samples were also taken from the same tuft of grass (e.g. A-1/A-2 and B-1/B-2 in different circles). **d** Number of loci with one (grey rectangle), two (orange rectangle) or three (black rectangle) different haplotypes. 'C' indicates *S. ciliata* circles, 'U' denotes *S. uniplumis* circles. Only the first sample ('−1') of each circle is labelled below the graph. The black dashed line separates putative tetraploid samples (with more than 1000 loci with three haplotypes) and diploid samples (with fewer than 1000 loci with three haplotypes). **e** Distributions of alternative-allele frequencies across all circles (C1 to C5 for *S. ciliata*, U1 to U15 for *S. uniplumis*) and the three paired samples from the same tufts of *S. uniplumis* (U-A to U-C). The blue trace refers to alternative-allele frequency distribution for sample 1, the purple trace for sample 2 and the green trace for sample 3 from the indicated circles; orange and green dashed lines indicate the two samples from one tuft. The vertical dashed grey lines indicate frequencies of 0.25 and 0.5. Circles that had grasses with different levels of ploidy are indicated in orange font. Sample U-4-3 (green trace) is a putative octaploid, with an alternative-allele frequency peak close to 0.125.

ten-fold lower for the remaining samples. This large number of loci with more than two haplotypes suggests that these samples are from tetraploid individuals. This is supported by the frequency distribution of alternative alleles showing either a sharp peak at 0.5 for the presumed diploid samples or a broader distribution with a peak at 0.25 for the presumed tetraploid samples (Fig. 1e). For *S. uniplumis* we also identified a putative octaploid sample (U4-3), with an alternative-allele frequency peak at 0.125 (Fig. 1e). Consistent with our findings, the genus *Stipagrostis* is known to contain both diploid and tetraploid species[22]; however, resolving the identity and origin of the different ploidy levels in our samples will require more work. Of particular relevance for our study is the finding that in both locations we found circles for which grasses in our three samples differed in their ploidy levels (Fig. 1e), strongly suggesting that they are not clonal.

**Genetic diversity is high among peripheral grasses.** To test the relatedness within circles directly, we performed hierarchical clustering on the genotypes at all variant sites that had a coverage above 11 in all samples of a given species (Fig. 2a, b). To account for the presence of both diploid and tetraploid samples in the populations, we recoded genotypes using relaxed settings, i.e. every position with more than 5% alternative or more than 5% reference calls was considered as heterozygous. This was done because the presence of homoeologous loci in the tetraploids means that reads will be sampled from four haplotypes rather than only two in the diploid samples. This in turn is expected to introduce more stochastic variation into the numbers of reads for each of the haplotypes. However, the tetraploid nature of many samples also means that many of the 'heterozygous' sites in these samples may in fact represent fixed differences between homoeologous loci, rather than heterozygosity of alleles of one locus. It is also for this reason, i.e. the presence of diploid and tetraploid samples, that we estimated genetic relatedness by hierarchical clustering rather than phylogenetic approaches.

The hierarchical clustering clearly separated the diploid and tetraploid samples. More importantly, it indicated that for both *S. uniplumis* and *S. ciliata* the samples from within one circle generally did not cluster together tightly but were interspersed with samples from other circles (Fig. 2a, b). Samples collected next to each other ('1' and '2') were not systematically more similar to each other than to sample '3' from the same circle (i.e. these samples are not ramets).

Of the three *S. uniplumis* sample pairs from single tufts, pair B-1 and B-2 was genetically identical, samples C-1 and C-2 were very closely related, yet samples A-1 and A-2 were clearly not genetically identical. This observation that the pair of samples A-1 and A-2 from the same tuft was not genetically identical was unexpected. This could be because of technical reasons, in

particular contamination with non-plant DNA and variation in genotype calls introduced by sampling from four haplotypes at homoeologous loci (all three pairs of samples from single tufts were tetraploid). To eliminate these potential sources of technical variation, we mapped the reads to the rice genome and retained only ones mapping in coding regions with a coverage of at least 11 reads across all samples of the respective species (*S. uniplumis* or *S. ciliata* separately). This therefore resulted in a stringent set of plant-specific loci with comparable coverage across all samples from the same species. Genotype calling and hierarchical clustering on this set indicated that pairs B-1 and B-2, as well as C-1 and C-2 were virtually identical (Fig. 3a, b), yet samples A-1 and A-2 were clearly more distinct from each other. Thus, our method can detect clonality of samples, yet even one tuft can consist of different genets. Importantly, the topology of the clustering based on the larger dataset (Fig. 2a, b) and the stringently filtered variants (Fig. 3a, c) was very similar, with the same assignment of samples to diploid and tetraploid clusters and similar intra-cluster relationships. With few exceptions among the tetraploid *S. uniplumis* samples (e.g. 2-1 and 2-2, 6-1 and 6-2, 11-1 and 11-2), samples from within a circle were not more closely related to each other than to samples from other circles, and neighbouring samples 1 and 2 within a circle were not more closely related to each other than to the third sample from the same circle.

To compare the degree of relatedness within and between circles quantitatively, we plotted the distributions of the minimal genetic distances within and between all combinations of circles (Fig. 4a). There was no statistically significant difference between the two distributions based on a Wilcoxon rank sum test using the *Oryza sativa* mapping-based genetic distances ($P = 0.34$), though the difference was significant when using stacks-based genetic distances ($P = 0.027$; Supplementary Fig. 2a). However, many between-circle distances were as small as the smallest within-circle distances (Fig. 4a), except for the identity seen within circle 2 (cf. Fig. 3a). Thus, our conclusion that individual circles do not form from a single centrifugally spreading genet and, therefore, are not clonal is robust to the analysis method chosen.

One concern with this interpretation could be that the FC periphery could be made up of different intermingled genets, comprising one centrifugally growing 'founding' genet and several other genets growing in from the surrounding matrix, but still respecting the bare centre of the circles. (Note that the distances between neighbouring FCs are generally very similar to the FC diameters at our sampling site for *S. uniplumis* [Supplementary Fig. 1].) If this was a frequent occurrence, we would expect to find ramets of the same invading genets from the matrix in the periphery of different neighbouring FCs. To test for

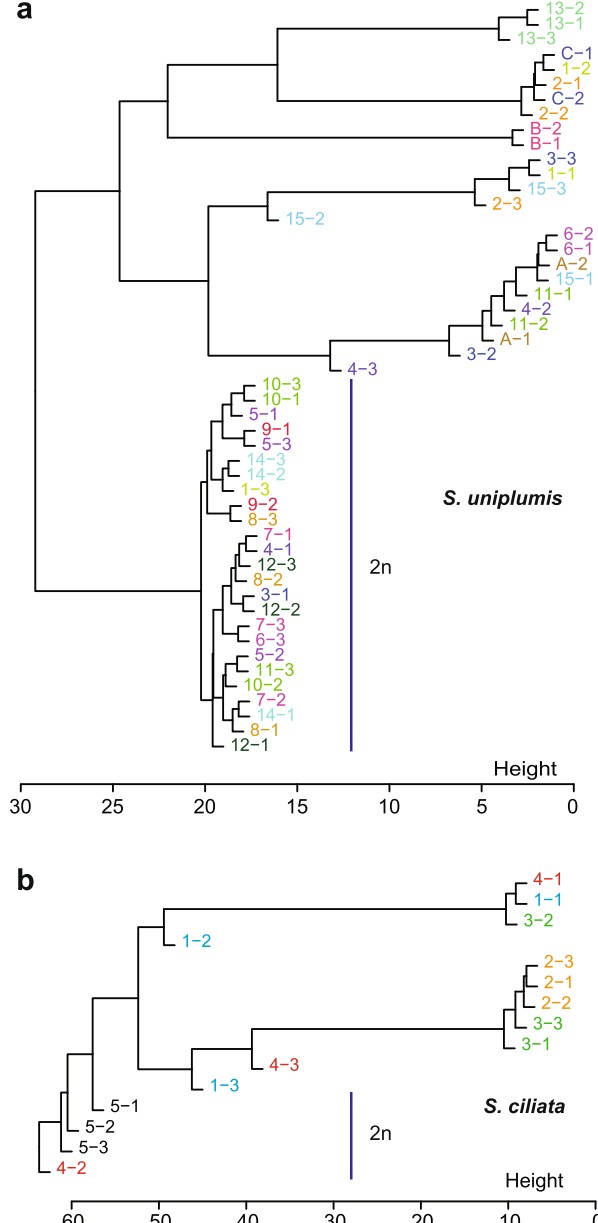

**Fig. 2 Pattern of genetic relatedness in Namib Desert fairy circles.**
**a** Hierarchical clustering of *S. uniplumis* samples across 15 circles, with three individuals per circle (e.g. 1-1, 1-2 and 1-3) and across three tufts of grass (A to C) with two samples per tuft (e.g. A-1 and A-2), based on genotypes at variant sites covered by more than 11 reads in each sample. Diploid samples are indicated (2n); the ones above are tetraploid, except for the octaploid sample 4-3. Samples from the same circle are indicated by the same font colour. **b** Hierarchical clustering of *S. ciliata* samples across 5 circles with three individuals per circle (e.g. 1-1, 1-2 and 1-3), based on genotypes at variant sites covered by more than 11 reads in each sample. Diploid samples are indicated (2n); the ones above are tetraploid. Samples from the same circle are indicated by the same font colour.

this possibility, we determined for each possible pair of FCs the minimal between-circle genetic distance and plotted this versus the spatial distance between the FC centres (Fig. 4b and Supplementary Fig. 2b). There was only a very weak relation between the minimal between-circle genetic distances and the spatial distances. Importantly, similar low genetic distances were found between neighbouring and between widely spaced FCs; these were of a similar magnitude to the low within-circle

distances. Fully excluding the above possibility of clonally growing 'founding' genets, intermingling with invading matrix genets would require much denser sampling of individual FCs. However, in our view the absence of any evidence for matrix clones spanning neighbouring FC peripheries clearly argues against a scenario of extensive clonal growth where genets from the matrix frequently intermingle and connect with nearby FC peripheries.

To test whether the above conclusions were sensitive to the choice of threshold for calling a sample heterozygous at a given position (5% used above) and to the frequency of alleles in the population, we re-analysed the *S. uniplumis* data based on either stacks or the rice-genome mapping by using a threshold of 10% for calling heterozygotes and requiring that all variant sites be present in at least two individuals. This should exclude artefacts due to PCR errors and duplicates. As shown in Supplementary Fig. 3, all our main conclusions (diploid vs. tetraploid samples; no clonality of all three samples from one circle; no closer relatedness of samples 1 and 2 from the same circle compared to sample 3; no relation between genetic and spatial distances) were also supported by the analyses with these altered parameters and filtering. We next sought to analyse the breeding pattern of the peripheral grasses. To do so, we focused on the diploid samples of *S. uniplumis* only, as the estimates of heterozygosity in the tetraploid samples are confounded by the fixed differences between homoeologous loci and as we only had four diploid samples of *S. ciliata*. We used the more conservative variant calls based on mappings against *O. sativa* coding regions to ensure plant-specific loci and minimize the influence of repeat sequences. Based on these data, we estimated the inbreeding coefficient $F$. The average value of $F$ across the diploid *S. uniplumis* samples was $-0.025$ (95% CI [$-0.048$, $-0.004$]), indicating that the plants were fully outbred.

## Discussion

Our results show that FCs formed by two different grass species in Namibia do not result from a vegetative growth pattern of a single clone. Rather, they are assembled from more than one genetically distinct genet, often of different ploidy levels, recruited from out-crossed seeds. This finding refutes the clonality hypothesis. As the circles are formed by unrelated genets, competition between genets within circles should dominate over competition between circles, because neighbours are an order of magnitude closer within (cm), than between (m), circles.

Our finding that, in the overwhelming majority of cases, peripheral grasses are not ramets of one genet indicates that FC peripheral individuals are unlikely to be the descendants of a single founder. This finding has implications for the relationship between the age of FCs and the lifespan of the grasses that surround them. *S. ciliata* and *S. uniplumis* are both facultative perennials[23]. Individuals flower in their first year if given limited moisture and then die. With extra moisture in their first season, after flowering they continue growth and produce rhizome buds. Then they die-back above-ground in the ensuing dry season and become dormant (see photographs of the same FCs before and after rain[9,11]). These individuals can then resprout from rhizome buds in the following wet season if that too is moist enough[23], otherwise they die completely. Given the slow decay rates in arid environments, dead *Stipagrostis* perennial plants are well known to persist in situ for many years and are commonly referred to as 'bloudak'[24]. Thus, it is only after prolonged droughts that the plants delimiting FCs almost disappear (e.g. Figure 1 in ref. [11]). The point is that the bare centres of FCs persist in situ almost indefinitely, despite being delimited by a periphery of short-lived live and persistent dead plants.

After several years of drought, the peripheral grasses emerge *de novo* in the landscape again in situ, given sufficient rain. Being facultative perennials, they are short-lived plants because their initial resource allocation is to flowering in their first year and

setting-seed, rather than to growth, which is in contrast to biennial or perennial growth forms. For example, Zimmerman et al. recorded 30% mortality of adult *S. uniplumis* perennials in a non-drought year, implying a mean age of less than 5 years in

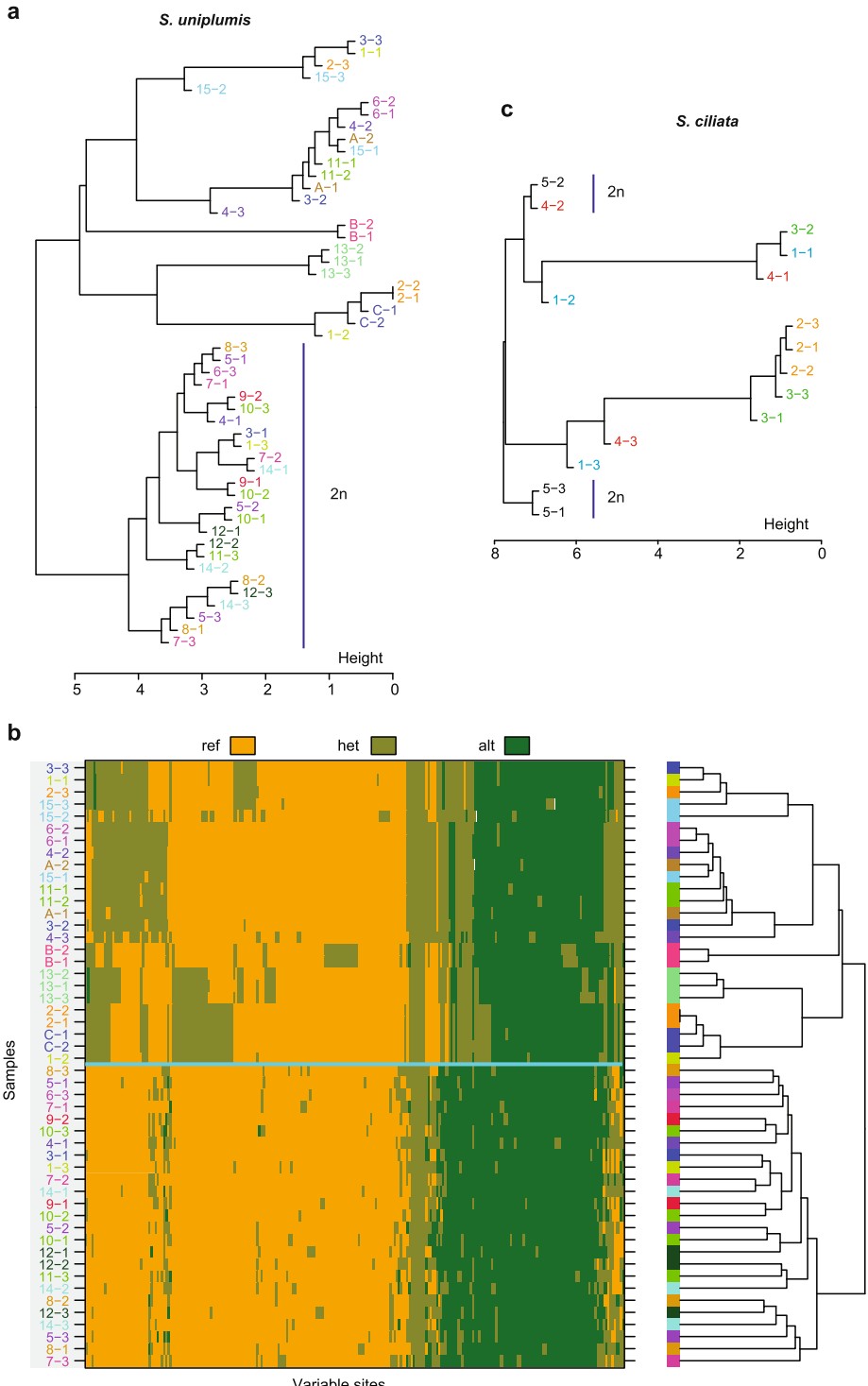

**Fig. 3 Genetic relatedness in fairy circles based on conserved plant-specific loci. a, c** Hierarchical clustering of *S. uniplumis* (**a**) and *S. ciliata* (**c**) samples based on recoded genotypes at variant sites after mapping reads to the *Oryza sativa* genome with coverage above 11 for all samples. Samples from the same circle are indicated by same font colour. Diploid samples are indicated (2*n*); the others are tetraploid, except for the octaploid 4-3 for *S. uniplumis*, with three individuals per circle (e.g. 1-1, 1-2 and 1-3) and across three tufts of grass (A to C) with two samples per tuft (e.g. A-1, A-2). **b** Heatmap with hierarchical clustering of *S. uniplumis* samples based on recoded genotypes at variant sites after mapping reads to the *O. sativa* genome with coverage above 11 for all *S. uniplumis* samples. Genotype calls at the sites are shown by colour (orange, olive, green) as defined at the top. Diploid samples are located below the turquoise line; the ones above are tetraploid, except for the octaploid 4-3. Samples from the same circle are indicated by the same font colour.

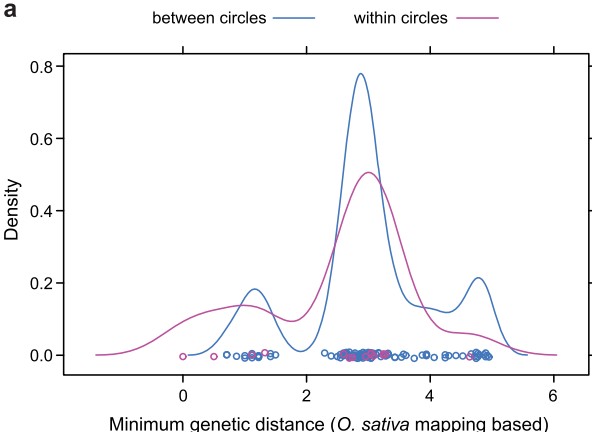

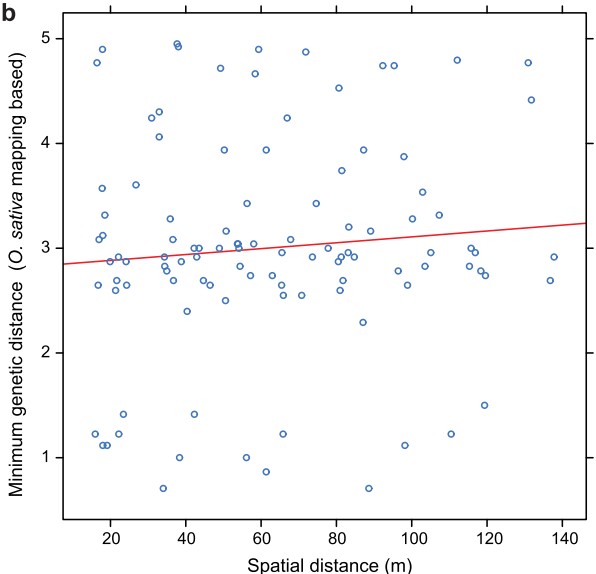

**Fig. 4 Relationship of within- and between-circle genetic distances based on mappings against the *O. sativa* genome. a** Density distributions of minimal genetic distances between samples from within the 15 *S. uniplumis* circles (purple; $n = 15$ within-circle distances) and between all possible pairs of circles (blue; $n = 105$ between-circle distances) are shown. Small circles indicate individual distances. **b** For each pair of *S. uniplumis* FCs the minimal genetic distance between samples from the two FCs is plotted relative to the spatial distance between the FCs. No within-circle comparisons are shown. The regression line is shown in red.

moist years[25]. Milton and Dean found mass mortality of *S. ciliata* during a drought with less than 15% of marked plants reaching 7 years of age[24]. They and Zimmerman et al. found mass seedling recruitment in wet years[26]. Thus, populations of these perennials are essentially single-aged cohorts, not clones, that establish in situ in two or more consecutive wet years and are then eliminated in decadal-level droughts[24]. There is thus a large mismatch in longevities of the position of FCs (bare centres lasting centuries) and that of individual peripheral non-clonal plants, which delimit the bare centres of FCs (decades). After drought mortality has eliminated most of the individuals at a site, it is unlikely that biotic interactions alone, such as competition between near neighbours, of the next cohort of short-lived non-clonal individuals, would be able to create a circle. It is similarly unlikely that the circles in a landscape are similarly sized and form in exactly the same place as previous circles, instantaneously (as circles form rapidly after sufficient rain), and that this process

would occur repeatedly in exactly the same place, with the same size and spatial pattern after several drought and wet cycles over a period of a century or more. The above mismatch is also a challenge to some of the VSO hypotheses. For example, Zelnik et al. suggest that births and deaths of FCs are gradual and that the FC spatial pattern emerges only after many years of interactions, after repeated droughts and spates[18]. This does not fit the observed rapid, *de novo*, in situ seedling establishment of FCs. In conclusion, our focus on the non-clonal, relatively short-lived life cycle of FC peripheral grass plants has implications for understanding the genesis, persistence and spatial patterns of FCs.

## Methods

**Plant material and sampling**. *S. uniplumis* (15 circles) and *S. ciliata* (5 circles) perimeter grasses were sampled at 19°05′31.58″S, 13°16′44.54″E and 24° 56′ 32.87″ S, 16° 01′ 12.82″E sites, respectively (Fig. 1a), under Permit 1854 (Ministry of Environment and Tourism, Namibia) and Permit NRNR/P/2014/01 (NamibRand Nature Reserve). FCs were at least 10 m apart. Grass samples were stored in silica in individual packets. The 15 *S. uniplumis* FCs were sampled from approximately 1 hectare (Supplementary Fig. 1). The sampling scheme is illustrated in Fig. 1c. Two of the three plants sampled per circle ('1' and '2') were located within 2 m of each other along the circumference of the FC periphery, while the third ('3') was sampled from the opposite side of the circle. For *S. uniplumis*, we also took two samples each from one tuft in three unrelated circles ('A' to 'C') to test for genetic relatedness within a tuft.

**ddRAD library preparation and sequencing**. 1 µg of DNA per sample was used for preparing the ddRAD library. Preparation of the library for ddRAD-seq was performed as described[27], using *Eco*RI and *Msp*I as restriction enzymes. The libraries were sequenced on a NextSeq 500 instrument with a mid-output kit in 2 × 75 paired-end mode.

**Data analysis**. Illumina paired-end sequencing data were demultiplexed using the stacks process_radtags program, version 1.41 for *S. ciliata* samples, version 2.0b for *S. uniplumis*. We checked for the presence of the *Eco*RI cut site using the -e option. Data were further processed using stacks version 2.41 (ref. [28]) with the following workflow: ustacks, cstacks, sstacks, tsv2bam and gstacks. Programs were run separately with default parameters. The number of haplotypes per tag was counted based on read 1 (R1) catalogue matches computed by sstacks for each sample. Allele frequency distributions were computed based on SNP genotype calls by gstacks based on the combined reads 1 and 2 for sites with a coverage of at least 11 reads excluding frequencies below 0.05 and above 0.95. To account for the polyploidy of the samples, genotype calls at variant sites were recoded as 0 (only reference allele is present), 0.5 (allele frequency above 5% for both alleles) and 1 (only alternative allele). Samples were grouped using hierarchical clustering of Euclidean distances between the recoded calls, including only sites with read depth above 11 in all samples analysed together (*S. uniplumis* and *S. ciliata* separately). Additionally, to exclude possible contaminants and to restrict the analysis to more conserved genome parts, reads were mapped against the *O. sativa* reference genome (Ensembl Plants, release 44. http://plants.ensembl.org) using BWA-MEM[29]. Mappings were further processed using samtools[30]. Variant calling within rice coding sequence regions was done using BCFtools[31,32] (http://samtools.github.io/bcftools/call-m.pdf). Genotype calls were recoded and grouped as with stacks. Inbreeding coefficients were calculated for diploid *S. uniplumis* samples using VCFtools[33].

Data processing, analysis and visualizations were done using R (R Core Team, 2017. https://www.R-project.org/) and R/lattice (http://lmdvr.r-forge.r-project.org). Genetic distances were computed as Euclidean distances between genotype calls of individual samples. Hierarchical clustering of those distances was done using the hclust function. Spatial distances between circles were calculated as distances between circle centres. Minimum genetic distances between circles were calculated as the minimum genetic distance between samples of the two circles. Minimum genetic distances within circles were calculated as the minimum genetic distance between any two samples within a circle. Differences between minimum genetic distances within and between circles were assessed by a Wilcoxon rank sum test using the wilcoxon.test function.

**Statistics and reproducibility**. This study concerns testing for genetic differences among three individual peripheral grasses on 15 separate FCs in one area and five FCs in another area to determine whether FCs were composed of genetically identical individuals (clones) or not. Although differences among just two individuals (the replicates in this study) per FC would confirm lack of clonality, we tested three individuals to strengthen the statistical results. Genetic differences among replicates and among FCs were determined statistically as differences in Euclidean distances between genotype calls of replicates by a Wilcoxon rank sum test.

**Reporting Summary**. Further information on research design is available in the Nature Research Reporting Summary linked to this article.

## Data availability
ddRAD sequencing data are available at NCBI SRA under accession number PRJNA576806. All raw data underlying the individual figures are provided as Supplementary Data.

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

## Acknowledgements
We are grateful to Michaela Preick and Michael Hofreiter for Illumina sequencing. This research was supported with funding from the University of Cape Town (JJM and NI), the Oppenheimer Trust (JJM), the University of Potsdam (ML) and a National Geographic Committee for Research and Exploration Grant (NB).

## Author contributions
N.I., J.J.M. and M.L. designed the study. N.B., M.C. and J.J.M. collected samples in the field. C.K. performed bioinformatics analysis of ddRAD-seq data. C.N.H. prepared ddRAD-seq libraries. M.L. drafted the manuscript with input from all authors, and all authors revised it for accuracy and content. All authors agreed to be held accountable for the content and approve the final version of the manuscript.

## Competing interests
The authors declare no competing interests.
