## [Peer Review File · Communications Biology]

Reviewers' comments:

Reviewer #1 (Remarks to the Author):

The manuscript 'Assembly of fairy circles in the Namib from genetically distinct, non-clonal plants' by Kappel et al. provides a well-written description of the results and implications of a genetic study of fairy circles (FCs) in two different regions in Namibia. Double-digest RADseq was used to genotype samples of *Stipagrostis uniplumis* and *S. ciliata* and test for clonality within FCs. The aim was to determine whether clonal expansion could explain the development of FCs which are characterised by a peripheral ring of plants surrounding a bare central area, a pattern consistent with clonality. As 2n and 4n individuals are known in both species, allele frequency was used to address possible ploidy variation in the sample set. This work makes a useful contribution to the literature investigating the formation of FCs because no genetic information is currently available and results clearly demonstrate the value of ddRADseq for identifying genetic individuals and assessing similarity. Listed below are some editorial suggestions and questions to be addressed by the authors.

1. Supplementary Fig 2: this seems to be missing (referred to in line 257, 262).
2. Getzin et al 2019 (Ecosphere 10, e02620) provide a landscape approach to studying FCs in Australia. It is worth citing as an interesting counterpart to the genetic approach of the current study.
3. I would like to see some additional information included in the main text: How long does it take a FC to form and how long do they last? Do the centres fill up with other vegetation at some point and a new FC form in the vicinity or the same place?
4. Given the number of FCs and their size, sampling of FCs, particularly of *S. ciliata*, and a small number of tufts in the peripheral rings of both species is rather low. What was the rationale behind sampling? While the results regarding clonality may not change, the small number of diploids sequenced has meant that the issue of ploidy (which was addressed well in analysis) has limited the use of the genetic results for investigating realised mating systems with results only available for *S. uniplumis*. It would be of value to models to confirm that sexual reproduction is effectively by outcrossing by comparing genetic similarity of plants from both species and at several locations. Please respond with additional explanation in main text.
5. Given the short length of the manuscript, I would like to see more of the Supplementary Discussion in the main text, either in the first section or under methods, unless word count does not permit an increase.
6. Line 21: insert comma after 'distribution'
7. Line 32: Is the 'matrix' the vegetation between different FCs?
8. Line 56: 100* = 100 times? In which case, spell out
9. Line 64: replace '>>' with 'much further'
10. Line 75: Insert 'along the circumference of the peripheral vegetation' after '... each other'
11. Line 77: How do you know the circles are unrelated before you test for that? How far apart were they? Were they different FCs from those 15 sampled for plants 1, 2 and 3?
12. Line 116: Sentence starting 'Also....' seems to have words missing. Suggest replacing 'suggests' with 'Our results suggest' and perhaps re-structuring for clarity.
13. Line 117: '...circles are formed of unrelated individuals....' Seems to contradict line 97 where it is stated that the 'C' pairs are closely related. It would be helpful to have the supplementary discussion about identifying 'A' pair as from the same genetic individual and 'C' pair as from another genetic individual whereas the 'B' pairs are not related at all despite their proximity in a single tuft (see comment 16, line 134 below)
14. Line 119-120: 'Explaining FC size and spatial patterns based on individual competition amongst unrelated plants within and between circles and the matrix, thus remains a challenge'. What do you propose as the next steps to help explain them?
15. Line 125: what was the minimum distance between FCs?
16. Line 134: Were any biological replicates included in sequencing to provide a baseline for biological variation (technical or biological) within a clone?
17. Line 141: delete 2nd 'based'
18. Line 142: insert comma after 'samples'
19. Line 145: delete 'so'; replace 'coverage' with 'read depth'
20. Line 237-240: re-structure or split into two sentences to improve clarity and flow.
21. Line 248: were raw ddRAD data run through a program such as Kraken (Wood and Slazberg

2014) to filter out microbial and fungal contaminants? This is standard practice to remove such contaminants.

22. Supplementary Fig 1c: Sample U-4-3 mentioned in caption is not identified in 1c (although present in Fig 1b of the main text). My apologies if I have mis-interpreted the plots. Do you mean that the left hand peak in U-4 is sample U-4-3?

23. 190: should 'Stipagrostis-Ciliata' be *Stipagrostis ciliata* (italicised)?

Reviewer #2 (Remarks to the Author):

Authors investigate very interesting natural phenomenon: rings of bare soil in arid grassland of Namibia and they are testing whether plant individual forming one ring are of clonal origin. There is couple of nice publications on this topic giving overview of hypotheses how those rings are formed and maintained. Among the hypotheses that rings of vegetation are result of clonal growth could be found (Danin & Orshan 1995 *J Arid Envir*; Getzin et al. 2015 *Ecography*). Presented manuscript is inspired by study done in Negev, Israel (Danin & Orshan 1995 *J Arid Envir*) where studied rings are around half of meter in diameter with rests of rhizomes inside of ring and are clearly result of clonal growth. For fairy rings in Namib desert, however van Rooyen et al. (2004 *J Arid Envir*) deny this possibility as rings in Namib desert are rather large (3-9m), small developmental stages are missing, and there is low amount of organic matter inside of the ring (van Rooyen et al. 2004 *J Arid Envir*). From this follows that results of presented manuscript that grasses forming rings are not clonally uniform around one ring and the ring is formed by different individuals and even ploidy levels is interesting and rule out one (although not very important) hypothesis of ring formation.

I miss discussion of other that genetic support for dismissing the hypothesis about clonal growth being responsible for ring formation. In arid climate one would expect that remnants of rhizomes may be visible in ring centers. Information about rhizome lateral spread of studied grasses is also missing. The only reference to clonal growth and lateral spread in other systems concerns one forest herb but more relevant literature is missing, for example: Bonanomi et al. 2014. Ring formation in clonal plants. *Community Ecology*. 15:77-86; Carteni et al. 2012. Negative plant coil feedback explaining ring formation in clonal plants. *J Theor Biol*. 313: 153-161; van Rooyen et al. 2004. Mysterious circles in the Namib Desert: review of hypotheses on their origin. *Journal of Arid Environments* 57: 467-485; Lanta et al. 2008. Radial growth and ring formation process in clonal plant *Eriophorum angustifolium* on post-mined peatland in the Sumava Mts., Czech Republic. *ANNALES BOTANICI FENNICI* 45: 44-54; Wikberg and Mucina 2002. Spatial variation in vegetation and abiotic factors related to the occurrence of a ring-forming sedge. *JOURNAL OF VEGETATION SCIENCE* 13: 677-684.

The title is misleading as plants forming circles are clonal grasses but circles were not formed by clonal growth

Conclusion: without more complex analyses of hypothesis on clonal growth origin of fairy rings including other aspects like details on clonal growth of studied plants, its lateral spread and direction, observations of remnants of rhizomes inside of circles, recording of moving of the ring periphery via clonal growth etc. the study is interesting anecdote.

Jitka Klimešová

Reviewer #3 (Remarks to the Author):

This study showed that two *Stipagrostis* species growing around the fairy circles in the arid-grasslands are mostly derived from seed reproduction, and some circles had individuals (ramets) with different polyploidy levels. This study contributes not only to the understanding of the formation of the fairy circle but also to the detection of clones with different polyploidy levels using genomic datasets (e.g. RAD-seq).

My major concern is about the sampling design. The authors found that non-clonal ramets (genetically distinct individuals) were found within a tuft (samples "A", "B", "C"; Lines 97, 101). This indicates that *Stipagrostis* genets could form highly intermingled patches, and indicates the results may not simply be due to seed reproduction (seed dispersal). Indeed, the evaluation of the magnitude of intermingling or segregation is known to be important for understanding the history of clonal growth and space occupation, and the competitive interactions among genets (Arnaud-Hond 2007 Molecular Ecology). If these species tend to form highly intermingled genets around the fairy circles, then the current sampling design (i.e. three samples per circle) would not be enough to rule out the clonal growth hypothesis (Line 100). High intermingling among genets may easily occur, for example, when the genets outside the fairy circle spread their distribution toward the edges of the circle, which happens even if the circles were actually created via clonal growth. This possibility could be taken into account if there is any prior knowledge on clonality of these species. Otherwise, evaluation of clonal structure of these species around and between the circles with high density sampling design may be necessary to support the conclusions made by the authors.

In addition, the authors did not provide sufficient methodological details in the data analysis section. Further description is necessary to reproduce their results and support their conclusions. Figure S2 was not found in the manuscript or files, therefore I did not make any comments regarding FigS2.

Other comments to the manuscript are as follows:

(L23) "individual fairy circles are unrelated individuals" should be rephrased since some samples were probably clones and not all of the samples tested in this study belong to unique genets.

(L32, 121) I found the term "matrix" difficult to understand. Additional explanation about "matrix" may be helpful.

(L64-67) The authors need to cite papers to support "1.2 m² yr⁻¹ biomass" is "unrealistically high population growth rate".^{[1][2]}

(L74) The term "individual" is confusing, use either ramets or genets, or define the term when first appeared in the manuscript.

(L75) The IDs for the samples within the circle ("1", "2", "3") is confusing when these appeared in the text or in the figure. Perhaps something like S1, S2, S3 or P1, P2, P3 could be more helpful for the readers.

(L78-82) FigS1b is showing the number of stacks (contigs), not tags. If my understanding is correct, the term "tags" (or rad tags) are used to distinct samples in RAD-seq, and the number of tags should be same as the number of samples. In addition, provide the mean and standard deviation of the number of stacks (contigs).

(L90-91) This should be rephrased since this study only investigated samples around the circles. The investigated samples were mostly not clonal, but this does not simply indicate the significance of seed dispersal.^{[1][2]}

(L103) There were some clones (genets) consisted with multiple ramets. Therefore "perennial grasses surrounding FCs are not clonal" should be rephrased.

(L105-107) I wonder why the authors did not use co-dominant markers such as SSR. SSR markers are often used to detect clones and handle different ploidy levels, and also able to handle much more samples since it requires much less cost for genotyping compared to RAD-seq. SSR markers can be designed from the short sequences such as from RAD-seq as well.

(L113-114) Again, "they were assembled from genetically distinct individuals" should be rephrased since some ramets were actually probably the same clones.

(L138) How did the authors process rad tags and conduct quality control? Did they use process_radtags program? Please specify.

(L139) Defining the optimized parameters of the stacks programs is important for the downstream analyses. The authors should specify the parameters they used to assemble and call variants in stacks programs (e.g. n, M), and the methods they used to optimize the parameters (e.g. r80 method). Did the authors use denovo_map.pl or did they use each program manually? Did the authors use "populations" program? These information are necessary to reproduce their results.

(L140) Provide reasons for only using read 1 (R1) but not using R2. Sequences were obtained in paired-end mode, so there should be both R1 and R2. Were variants in read 2 (R2) used in the detection of clones and in hierarchical clustering?^[17]_{SEP}

(L141) Change "based based on" to "based on".

(L142) What kind of variant sites included in this study? Did the authors include SNPs, indels, MNPs, etc? Please specify.

(L143) As described in the supplementary discussion, the threshold of 5% is probably very loose since RAD-seq are prone to PCR duplicates. It may be beneficial to show the results (figures) using more strict condition (e.g. 10%, 15%) as supplementary material.

(L144) Provide reasons for only assessing one species (*S. uniplumis*) and not assessing *S. ciliata*.

(L145) Did the authors retained variant sites with depth of coverage larger than 11 in "all" samples (samples in both locations)? Were any other filtering done in this study? For example, filtering based on minor allele frequency (MAF) is as important as filtering with depth of coverage to reduce the genotyping errors at the variant sites. In addition, the authors should specify the number of variants obtained in the stacks pipeline and how much were retained after the filtering in the final dataset.

(L154) If I am not mistaken, only data analysis done except for stacks program is probably hierarchical clustering. Then "Data analysis" should be changed to "hierarchical clustering" and the authors should specify which program (package) was used for the clustering.

(FigS1b) How did the authors obtain the threshold (the green dashed line) in Fig S1b? Numbers (1, 2, 3) in the figure legend is confusing since the authors are using the same numbers to represent the samples within the circle. Does "3" in the figure legend indicate haplotype count more than two?

Referee expertise:

Referee #1: Plant reproduction & genetics

Referee #2: Plant clonality and functional ecology

Referee #3: Plant ecology and genetics

Reviewers' comments:

Reviewer #1 (Remarks to the Author):

The manuscript 'Assembly of fairy circles in the Namib from genetically distinct, non-clonal plants' by Kappel et al. provides a well-written description of the results and implications of a genetic study of fairy circles (FCs) in two different regions in Namibia. Double-digest RADseq was used to genotype samples of *Stipagrostis uniplumis* and *S. ciliata* and test for clonality within FCs. The aim was to determine whether clonal expansion could explain the development of FCs which are characterised by a peripheral ring of plants surrounding a bare central area, a pattern consistent with clonality. As 2n and 4n individuals are known in both species, allele frequency was used to address possible ploidy variation in the sample set. This work makes a useful contribution to the literature investigating the formation of FCs because no genetic information is currently available and results clearly demonstrate the value of ddRADseq for identifying genetic individuals and assessing similarity. Listed below are some editorial suggestions and questions to be addressed by the authors.

We thank the reviewer for these positive comments.

1. Supplementary Fig 2: this seems to be missing (referred to in line 257, 262).

This was corrected after submission but the updated version of the manuscript was not submitted to some referees. The revised version includes all figures and supplementary figures. Note that the previous Supplemental Figure 2 is now Figure 3.

2. Getzin et al 2019 (Ecosphere 10, e02620) provide a landscape approach to studying FCs in Australia. It is worth citing as an interesting counterpart to the genetic approach of the current study.

We have expanded the introduction (lines 52 onward) to include reference to Getzin's research which rejects the termite hypothesis for Australian FCs..

3. I would like to see some additional information included in the main text: How long does it take a FC to form and how long do they last? Do the centres fill up with other vegetation at some point and a new FC form in the vicinity or the same place?

The introduction of the manuscript has been revised to include a discussion on these important points (lines 42-51) and we include citations to the following papers:

*van Rooyen, M. W., Theron, G. K., van Rooyen, N., Jankowitz, W. J. & Matthews, W. S. Mysterious circles in the Namib Desert: review of hypotheses on their origin. J Arid Environ 57, 467–485 (2004).
Tschinkel, W. R. The Life Cycle and Life Span of Namibian Fairy Circles. PLoS One 7, (2012).
Juergens, N. The Biological Underpinnings of Namib Desert Fairy Circles. Science 339, 1618–1621 (2013).*

Tschinkel (2012) in particular provides data on the birth, life and death of FC's. They are long-lived up to 75 years. We have cited data from this paper.

4. Given the number of FCs and their size, sampling of FCs, particularly of *S. ciliata*, and a small number of tufts in the peripheral rings of both species is rather low. What was the rationale behind sampling? While the results regarding clonality may not change, the small number of diploids sequenced has meant that the issue of ploidy (which was addressed well in analysis) has limited the use of the genetic results for investigating realised mating systems with results only available for *S. uniplumis*. It would be of value to models to confirm that sexual reproduction is effectively by outcrossing by comparing genetic similarity of plants from both species and at several locations. Please respond with additional explanation in main text.

The clonality hypothesis can be rejected if just two individuals within a circle are not clones so our use of 3 individuals has some redundancy. Similarly, our results of no clonality have implications for the vegetation self-organisation hypothesis. Although our study was not directed at breeding-systems it was clear from the results that out-breeding and seed dispersal were significant and thus we commented on this.

Testing the clonality hypothesis was the focus of this research and informed the design of the sampling. We were surprised to find that not only were plants in FCs not clonal, but that they were also a mixture of diploid and tetraploid species. Investigating the breeding systems and population genetics of the diploid and tetraploid species will require a different experimental design and will be the subject of future research.

Also, in response to reviewer 3 below we have included a spatial analysis, comparing minimal genetic distances between circles with their spatial distances (lines 190 onwards). This indicated that there was no clear correlation between genetic and spatial distances.

5. Given the short length of the manuscript, I would like to see more of the Supplementary Discussion in the main text, either in the first section or under methods, unless word count does not permit an increase.

The supplementary discussion has been moved to either the main text or materials and methods.

6. Line 21: insert comma after 'distribution'

This sentence in the abstract has been rephrased.

7. Line 32: Is the 'matrix' the vegetation between different FCs?

Yes, the matrix is the space and plants between FCs. The manuscript has been edited to define matrix (lines 37-38).

8. Line 56: 100* = 100 times? In which case, spell out

Thank you for picking up this typographical error, it has been corrected (line 88).

9. Line 64: replace '>>' with 'much further'

This has been corrected.

10. Line 75: Insert 'along the circumference of the peripheral vegetation' after '... each other'

The manuscript has been edited to incorporate this suggestion (lines 122-123).

11. Line 77: How do you know the circles are unrelated before you test for that? How far apart were they? Were they different FCs from those 15 sampled for plants 1, 2 and 3?

Gps points were taken at each circle, and this is now included in Supplementary Figure 1. Nearest neighbour circles are about 10 m apart. The samples taken from the same tufts were from circles distinct from the 15 others.

12. Line 116: Sentence starting 'Also... ' seems to have words missing. Suggest replacing 'suggests' with 'Our results suggest' and perhaps re-structuring for clarity.

The original sentence

"Also, given that we have shown the circles are formed of unrelated individuals, suggests that competition between these individuals should dominate over competition between circles, because near neighbours are an order of magnitude closer within, than between, circles."

Has been changed to

As the circles are formed by unrelated genets, competition between genets within circles should dominate over competition between circles, because neighbours are an order of magnitude closer within (cm), than between (m), circles. (lines 229-231)

13. Line 117: '...circles are formed of unrelated individuals...' Seems to contradict line 97 where it is stated that the 'C' pairs are closely related. It would be helpful to have the supplementary discussion about identifying 'A' pair as from the same genetic individual and 'C' pair as from another genetic individual whereas the 'B' pairs are not related at all despite their proximity in a single tuft (see comment 16, line 134 below)

We have moved the discussion about the A-C pairs into the main text (line 157 onwards line 121 ??). Material for each pair was sampled from leaves within a tuft of grass, rather than from tufts that are either 2m apart ("1", "2") or across the FC (sample "3"). Samples "1" "2" and "3" from the same FC were not related. We expected that the paired samples, ie A-1/A-2, B-1/B-2 and C-1/C-2 would be identical because the samples in each pair came from the same tuft. This does not contradict our findings, in fact, it strengthens our findings, that two individuals can even be found in the same tuft of grass.

14. Line 119-120: 'Explaining FC size and spatial patterns based on individual competition amongst unrelated plants within and between circles and the matrix, thus remains a challenge'. What do you propose as the next steps to help explain them?

*Our results indicate peripheral grasses are discrete individuals and given their small size (< 0.25 m tall for *S. uniplumis*), this implies a short lifespan. Many *Stipagrostis* species are facultative perennials; they flower in their first year and if conditions are good, such as on FC peripheries, they can perennate and sprout the following year but die-back completely in drought years. Given this short-lived life-history we predict that lifespans of the individual peripheral grasses are considerably less than the lifespans of individual FCs. If correct this implies that the remarkable spatial pattern is imposed on generations of individuals rather than individuals creating the pattern. We have spelled this out in more detail in the Discussion (line 232 onwards).*

15. Line 125: what was the minimum distance between FCs?

10 m. Please see the response to point 11 above. The materials and methods have been updated to include this information

16. Line 134: Were any biological replicates included in sequencing to provide a baseline for biological variation (technical or biological) within a clone?

*We presume the reviewer means technical replicates, ie did we independently analyse the same sample twice? This was not done. However, we did collect duplicate samples from three tufts (i.e. Samples UA-1/UA-2, UB-1/UB-2 and UC-1/UC-2 for *S. uniplumis*). The methods have been updated to make this more clear.*

17. Line 141: delete 2nd 'based'

Deleted.

18. Line 142: insert comma after 'samples'

inserted.

19. Line 145: delete 'so'; replace 'coverage' with 'read depth'

We have corrected the manuscript.

20. Line 237-240: re-structure or split into two sentences to improve clarity and flow.

This has been split into two sentences (lines 145-149).

21. Line 248: were raw ddRAD data run through a program such as Kraken (Wood and Slazberg 2014) to filter out microbial and fungal contaminants? This is standard practice to remove such contaminants.

*We did not filter out contaminants from raw fastq data. Yet as we only used sites with read depth above 11 in all samples analyzed together, only highly abundant contaminants in all of those would have affected our analysis. The *O. sativa* mapping based analysis should have completely excluded contaminants, as we only use reads mappings against coding sequences. As this mapping based analysis confirms the stacks based one, we did not opt for repeating the analysis with initial filtering.*

22. Supplementary Fig 1c: Sample U-4-3 mentioned in caption is not identified in 1c (although present in Fig 1b of the main text). My apologies if I have mis-interpreted the plots. Do you mean that the left hand peak in U-4 is sample U-4-3?

Yes, the green line for circle U4 is sample U4-3. We have added a legend for the relation of sub-circle samples to colours to the Figure (now Figure 1e). We have also explained this more clearly in the legend to Figure 1e.

23. 190: should 'Stipagrostis-Ciliata' be *Stipagrostis ciliata* (italicised)?

This has been corrected.

Reviewer #2 (Remarks to the Author):

Authors investigate very interesting natural phenomenon: rings of bare soil in arid grassland of Namibia and they are testing whether plant individual forming one ring are of clonal origin. There is couple of nice publications on this topic giving overview of hypotheses how those rings are formed and maintained. Among the hypotheses that rings of vegetation are result of clonal growth could be found (Danin & Orshan 1995 J Arid Envir; Getzin et al. 2015 Ecography). Presented manuscript is inspired by study done in Negev, Israel (Danin & Orshan 1995 J Arid Envir) where studied rings are around half of meter in diameter with rests of rhizomes inside of ring and are clearly result of clonal growth. For fairy rings in Namib desert, however van Rooyen et al. (2004 J Arid Envir) deny this possibility as rings in Namib desert are rather large (3-9m), small developmental stages are missing, and there is low amount of organic matter inside of the ring (van Rooyen et al. 2004 J Arid Envir). From this follows that results of presented manuscript that grasses forming rings are not clonally uniform around one ring and the ring is formed by different individuals and even ploidy levels is interesting and rule out one (although not very important) hypothesis of ring formation.

It is correct that van Rooyen et al 2004 J Arid Environ voiced the opinion that clonality was not important. However, this was an opinion; the clonality hypothesis can only be tested genetically, as we have done. Secondly, our demonstration of lack of clonality is important to the vegetation self-organisation hypothesis. Both the modelling approach and the parameters of this model are sensitive to individuality.

I miss discussion of other that genetic support for dismissing the hypothesis about clonal growth being responsible for ring formation. In arid climate one would expect that remnants of rhizomes may be visible in ring centers. Information about rhizome lateral spread of studied grasses is also missing. The only reference to clonal growth and lateral spread in other systems concerns one forest herb but more relevant literature is missing, for example: Bonanomi et al. 2014. Ring formation in clonal plants. Community Ecology. 15:77-86; Carteni et al. 2012. Negative plant coil feedback explaining ring formation in clonal plants. J Theor Biol. 313: 153-161; van Rooyen et al. 2004. Mysterious circles in the Namib Desert: review of hypotheses on their origin. Journal of Arid Environments 57: 467-485; Lanta et al. 2008. Radial growth and ring formation process in clonal plant *Eriophorum angustifolium* on post-mined peatland in the Sumava Mts., Czech Republic. ANNALES BOTANICI FENNICI 45:44-54; Wikberg and Mucina 2002. Spatial variation in vegetation and abiotic factors related to the occurrence of a ring-forming sedge. JOURNAL OF VEGETATION SCIENCE 13: 677-684.

These papers, noted by reviewer2 shows that clonality is widespread in nature and worth taking seriously. We have expanded the introduction to the manuscript to include several of these papers. Some aspects of the clonality hypothesis cannot easily be tested for Fairy Circles because growth is sporadic in this highly variable very arid environment; it only occurs in wet years. For example, the lack of remains of rhizomes may just reflect previous droughts

The title is misleading as plants forming circles are clonal grasses but circles were not formed by clonal growth

No, the point of our paper is that in these environments these are not clonal grasses.

Conclusion: without more complex analyses of hypothesis on clonal growth origin of fairy rings including other aspects like details on clonal growth of studied plants, its lateral spread and direction, observations of remnants of rhizomes inside of circles, recording of moving of the ring periphery via

clonal growth etc. the study is interesting anecdote.

Firstly, we have shown that near neighbour individuals are not clones. Therefore, there is no need to look for rhizomes etc. Secondly, our paper also has relevance to the self-organisation hypothesis and not just the clonality hypothesis. We have expanded on these points in the Discussion (lines 225 onwards).

Jitka Klimešová

Reviewer #3 (Remarks to the Author):

This study showed that two *Stipagrostis* species growing around the fairy circles in the arid-grasslands are mostly derived from seed reproduction, and some circles had individuals (ramets) with different ploidy levels. This study contributes not only to the understanding of the formation of the fairy circle but also to the detection of clones with different ploidy levels using genomic datasets (e.g. RAD-seq).

My major concern is about the sampling design. The authors found that non-clonal ramets (genetically distinct individuals) were found within a tuft (samples "A", "B", "C"; Lines 97, 101). This indicates that *Stipagrostis* genets could form highly intermingled patches, and indicates the results may not simply be due to seed reproduction (seed dispersal). Indeed, the evaluation of the magnitude of intermingling or segregation is known to be important for understanding the history of clonal growth and space occupation, and the competitive interactions among genets (Arnaud-Hound 2007 Molecular Ecology). If these species tend to form highly intermingled genets around the fairy circles, then the current sampling design (i.e. three samples per circle) would not be enough to rule out the clonal growth hypothesis (Line 100). High intermingling among genets may easily occur, for example, when the genets outside the fairy circle spread their distribution toward the edges of the circle, which happens even if the circles were actually created via clonal growth. This possibility could be taken into account if there is any prior knowledge on clonality of these species. Otherwise, evaluation of clonal structure of these species around and between the circles with high density sampling design may be necessary to support the conclusions made by the authors.

This paper set out to test the hypothesis that all grasses in the periphery of a fairy circle were genetically identical. Based on our results we feel confident in rejecting this hypothesis. That said, we agree that the suggested possibility of highly intermingled genets is conceivable. In particular, we understand the reviewer's comment as suggesting that circles might be formed by a centrifugally growing 'founding' genet, but that other clonally growing genets from the matrix could invade the periphery, resulting in non-clonality of the periphery. If this were a common occurrence, we should find ramets from the same 'matrix genets' to be present in the peripheries of neighbouring circles. This is a notion we have tested in Figure 4b and Supplementary Figure 2b, where we plot the minimal between-circle genetic distance for each pair of circles relative to the spatial distance between circles. In contrast to the above prediction, between-circle genetic distances were not related to spatial distances, and very similar samples could be found between nearest-neighbour circles and between distant circles with similar frequency. Thus, next to the 'pure' clonal growth hypothesis (all peripheral grasses in a FC representing one genet) we also consider the intermingled clonal growth hypothesis suggested by the reviewer (extensive clonal growth in the matrix obscuring a clonal signal within the circles) as unlikely. We acknowledge that fully excluding this possibility would require much denser sampling of individual circles. A discussion of these aspects has been added (lines 190-206).

In addition, the authors did not provide sufficient methodological details in the data analysis section. Further description is necessary to reproduce their results and support their conclusions. Figure S2 was not found in the manuscript or files, therefore I did not make any comments regarding FigS2.

We have added a diagram to illustrate the sampling method more clearly (Figure 1c) and expanded the Methods section (see more details below).

A corrected version of the manuscript, including a correctly formatted FigS1 and the missing FigS2 was loaded to Communications Biology on March 31, but unfortunately was not distributed to the reviewers. The revised manuscript includes all figures.

Other comments to the manuscript are as follows:

(L23) “individual fairy circles are unrelated individuals” should be rephrased since some samples were probably clones and not all of the samples tested in this study belong to unique genets.

*The only circle where all three samples could represent the same clone is circle 2 of *S. ciliata*. In all other circles at least one of the three peripheral samples was genetically distinct. We have therefore changed the sentence to “Here, we show that for virtually all tested fairy circles the periphery is not exclusively made up of genetically identical grasses, but these peripheral grasses belong to more than one unrelated genet.” (lines 24-26).*

(L32, 121) I found the term “matrix” difficult to understand. Additional explanation about “matrix” may be helpful.

Matrix is the term previous authors have used for the space and plants between fairy circles. We now define this more clearly in the text

*“The circles are typically made up of perennial tufts of *Stipagrostis* grass growing around barren central patches roughly 2-10 m in diameter (Figure 1a), with annual or perennial grasses in the area between circles; the matrix.”*

(L64-67) The authors need to cite papers to support “1.2 m² yr⁻¹ biomass” is “unrealistically high population growth rate”.

We have added an explanation for why this is a unrealistically high growth rate (lines 93-102).

(L74) The term “individual” is confusing, use either ramets or genets, or define the term when first appeared in the manuscript.

Our paper was testing whether tufts of grass in a FC are clonal (ie ramets) or genetic individuals (genets). We have clarified this in the text and checked that the terms ramet and genet are used consistently throughout the manuscript, with a definition provided in lines 62-63.

(L75) The IDs for the samples within the circle (“1”, “2”, “3”) is confusing when these appeared in the text or in the figure. Perhaps something like S1, S2, S3 or P1, P2, P3 could be more helpful for the readers.

We used (“1”, “2”, “3”) (tufts from adjacent individuals and an opposite tuft from one circle) and (‘A’ to ‘C’) (pairs of samples from the same tuft) to define the sampling of tufts of grass in a fairy circle. We have clarified this by adding Figure 1c and explaining the sample labelling in the legend to Figure

1c.

(L78-82) FigS1b is showing the number of stacks (contigs), not tags. If my understanding is correct, the term “tags” (or rad tags) are used to distinct samples in RAD-seq, and the number of tags should be same as the number of samples. In addition, provide the mean and standard deviation of the number of stacks (contigs).

The term ‘tags’ is used for the different individual fragments obtained by RAD-seq from one sample. We changed tags to loci to follow the stacks terminology. Average loci/contig count is 63096 (sd=15436), this was added to the manuscript (line 127).

(L90-91) This should be rephrased since this study only investigated samples around the circles. The investigated samples were mostly not clonal, but this does not simply indicate the significance of seed dispersal.

We have deleted the reference to seed dispersal.

(L103) There were some clones (genets) consisted with multiple ramets. Therefore “perennial grasses surrounding FCs are not clonal” should be rephrased.

Following the inclusion of the previous Supplemental Discussion in the main text (following reviewer 1’s suggestion), we have rewritten this paragraph. It now ends on the statement “Thus, our conclusion that individual circles do not form from a single centrifugally spreading genet and are therefore not clonal is robust to the analysis method chosen.” (lines 188-189)

(L105-107) I wonder why the authors did not use co-dominant markers such as SSR. SSR markers are often used to detect clones and handle different ploidy levels, and also able to handle much more samples since it requires much less cost for genotyping compared to RAD-seq. SSR markers can be designed from the short sequences such as from RAD-seq as well.

We did not have any information at hand to design primers for microsatellite markers (SSR). We initially tried using ISSRs, which does not require prior knowledge of a genome’s sequence, but found that the method was not reliable, and did not give sufficient resolution of PCR products to test our hypothesis accurately. This is why we chose to apply DDrad-seq to the problem as this method is not dependent on having prior gene sequence information. The reviewer is correct that we could use the information from the DDrad-seq assemblies to design primers for SSRs for future studies.

(L113-114) Again, “they were assembled from genetically distinct individuals” should be rephrased since some ramets were actually probably the same clones.

This sentence has been rephrased to read:

“Rather, they are assembled from more than one genetically distinct genets, often of different ploidy levels, recruited from out-crossed seeds.”

(L138) How did the authors process rad tags and conduct quality control? Did they use process_radtags program? Please specify.

We used process_radtags for demultiplexing Illumina sequencing data. We checked for the presence of the EcoRI cut site using the -e option. This has been added to the Methods section (lines 292-293).

(L139) Defining the optimized parameters of the stacks programs is important for the downstream analyses. The authors should specify the parameters they used to assemble and call variants in stacks programs (e.g. n, M), and the methods they used to optimize the parameters (e.g. r80 method). Did the authors use denovo_map.pl or did they use each program manually? Did the authors use “populations” program? These information are necessary to reproduce their results.

Sub programs of stacks were used separately, this has been clarified in the Methods section (line 294). The populations program was not used for the work presented here. We also did analyses with stacks version 1 optimizing/testing multiple parameters. Those analyses did not lead to different results, and are not presented in the manuscript. Importantly, the orthogonal O. sativa mapping based analysis fully supports the stacks based results, giving us confidence in their validity.

(L140) Provide reasons for only using read 1 (R1) but not using R2. Sequences were obtained in paired-end mode, so there should be both R1 and R2. Were variants in read 2 (R2) used in the detection of clones and in hierarchical clustering?

We directly used the output of sstacks for this haplotype analysis (.matches.txt.gz files). The stacks pipeline only integrates the second read pair with the tsv2bam step for use in gstacks. To our opinion this is sufficient to show the presence of different ploidy levels. We used the gstacks variant calling including both read pairs for all further analyses. This has been clarified in the Methods (lines 295-299).

(L141) Change “based based on” to “based on”.

This edit has been made.

(L142) What kind of variant sites included in this study? Did the authors include SNPs, indels, MNPs, etc? Please specify.

We restricted our analysis to SNPs and have added this information to the Data analysis section. (line 297)

(L143) As described in the supplementary discussion, the threshold of 5% is probably very loose since RAD-seq are prone to PCR duplicates. It may be beneficial to show the results (figures) using more strict condition (e.g. 10%, 15%) as supplementary material.

Please see the response to the comment on L145 below.

(L144) Provide reasons for only assessing one species (*S. uniplumis*) and not assessing *S. ciliata*.

*We have performed the hierarchical clustering also for *S. ciliata* (Figures 2b and 3b).*

(L145) Did the authors retained variant sites with depth of coverage larger than 11 in “all” samples (samples in both locations)? Were any other filtering done in this study? For example, filtering based on minor allele frequency (MAF) is as important as filtering with depth of coverage to reduce the genotyping errors at the variant sites. In addition, the authors should specify the number of variants obtained in the stacks pipeline and how much were retained after the filtering in the final dataset.

*We used the above 11X depth threshold at the level of all samples to be analyzed together (so *S. unipluis* and *S. ciliata* separately), and have now clarified this in the manuscript (lines 142, 168-169 and line 303).*

*To test whether our conclusions were sensitive to misgenotyping because of PCR errors and duplicates (see also your comment on L143 above), we repeated the analyses for *S. uniplumis* using a 10% alternative-allele threshold for recoding genotypes as heterozygous and required that variant sites were found in at least two individuals. The results of the hierarchical clustering and the comparison of genetic versus spatial distances are shown in Figure S3 and described in lines 207-215). While some minor aspects of the topology of the trees were changed, for example because of the lower power to detect heterozygotes in the tetraploid samples, all of our main conclusions (diploid vs. tetraploid samples; no clonality of all three samples from one circle; no closer relatedness of samples 1 and 2 from the same circle compared to sample 3; no relation between genetic and spatial distances) were still fully supported by this analysis.*

(L154) If I am not mistaken, only data analysis done except for stacks program is probably hierarchical clustering. Then “Data analysis” should be changed to “hierarchical clustering” and the authors should specify which program (package) was used for the clustering.

*We have also performed the analysis independently of the stacks program, using mapping of the reads to the *O. sativa* reference genome. Therefore, we have retained the title of the Methods section as ‘Data analysis’. Hierarchical clustering was done using the hclust function from core R, as is now explained in the Methods (line 314).*

(FigS1b) How did the authors obtain the threshold (the green dashed line) in Fig S1b? Numbers (1, 2, 3) in the figure legend is confusing since the authors are using the same numbers to represent the samples within the circle. Does “3” in the figure legend indicate haplotype count more than two?

The line in the figure (now Figure 1d) was drawn based on the visual separation of the samples according to the frequency of stacks with three haplotypes and on the results from the allele-frequency plots in Figure 1e. We have explained the labelling more clearly in the legend. ‘3’ indicates stack with three haplotypes; stacks with more than three haplotypes are very few and not shown.

REVIEWERS' COMMENTS:

Reviewer #1 (Remarks to the Author):

The revised paper has addressed my previous main concern although one option that has not been raised is the possibility that the bare circles are the result of fungal growth. A fungus could gradually expand and prevent re-establishment of grasses within the circle for a number of reasons including nutrients or allelopathy. The association between mycorrhizal fungi and grasses is well-known, sometimes with the effect of exhausting nutrients as the growth expands from a central point whilst providing higher nutrient availability at the leading edge around the periphery, or producing chemicals that prevent plant growth. Nutrition could explain higher growth of plants obvious from your images at the edge of the FCs and allelopathy could provide an explanation of why the centre is not re-colonised. A fungal influence would also fit with the findings of this study that there were different genets making up a FC despite the overall pattern of the FC being typical of clonal growth. If this option has not been explored for FCs (I couldn't find any literature), it would be useful to mention it as a potentially fruitful area for further FC research. There are a few very minor editorial comments that need to be addressed such as: 1. Be consistent with use of et al or et al. in text (e.g. line 78 vs 106); 2. Ensure space after abbreviation of genus (e.g. line 161).

Reviewer #2 (Remarks to the Author):

I am generally satisfied with changes and context provided by authors concerning clonal growth of studied species and system – the story now is much better framed. I have still three points that I think need to be elucidated:

(1) Page 5: „These literature values may be unrealistic for FCs. For example, 1.2 m² yr⁻¹ is likely to be too high for clonal growth in the arid circumstances in which fairy circles occur. *Stipagrostis* individual plants across the landscape are typically only 0.005-0.13 m² in canopy area with a mean area of 0.05 m² (ref. 20). Clonal spread in other systems can similarly also be orders of magnitude lower than 1.2 m² yr⁻¹ (ref. 19)“

My comment: How you are assessing the canopy area of *Stipagrostis* individuals? The „individuals“ may be ramets connected by rhizome therefore their individual area without knowing whether they belong to one clone (and this is what you want to find in your study) is not important in this respect. Consider reformulating this part as follows: „These literature values may be unrealistic for FCs. For example, 1.2 m² yr⁻¹ is likely to be too high for clonal growth in the arid circumstances where it might be orders of magnitude lower.“

Page 5: „Finally, if the assumed 1.2 m² yr⁻¹ biomass spread¹⁸ is due to total growth of new recruits per parent individual, then this implies unrealistically high population growth rates (>20 new recruits, each with 0.05 m² in canopy volume²⁰ required per parent individual).“

My comment: I do not understand an idea behind this text: How those rings may be formed by recruitment from seeds? Production of generative offspring is not biomass spread... they are not connected to share water... what if “biomass spread” is due to production of ephemeral roots toward the center of the ring and preempting the water resource. The species growing around the patch may be much better in production of such ephemeral roots than matrix species and so they gain fitness advantage after rain and it allows them to keep position along the ring – where they survive better as they are sheltered by matrix species during their germination and probably also later. It is only my speculation, but what if those ephemeral roots may be observed only short time after rain similarly as plants in snow-beds are producing ephemeral roots growing to snowpack? (see Onipchenko et al. 2009 Ecological Letters)

(2) Page 10, second paragraph and page 11 of the discussion. Here we are informed that plants forming rings and are subject of the study are short-lived perennials (living 5 - 7 years). Still the whole study is motivated to find out whether those grasses form rings 2-10 m in diameter by clonal growth. Such clonal rings would therefore need lateral spread of the clone 20 – 100 cm per year! Such lateral spread is highly improbable in arid environment! Therefore, as the information is from existing literature it must be cited already in the introduction and it will be argument against theory that the rings have clonal origin.

Page 10: Change “...if it too is moist enough” to “...if it is moist enough”

Page 11: “Being facultative perennials, they are short-lived plants because their initial resource

allocation is to flowering in their first year and setting-seed, rather than to growth, which is in contrast to biennial or perennial growth forms.”

My comment: it is not clear what exactly is in contrast with biennial and perennial growth forms as they differ very much in their allocation (and perennial herbs often flower first year of their life if in good conditions...)

(3) Already during first round or reviews I asked for the change in the title. The title “The assembly of fairy circles in Namibia is from genetically distinct, non-clonal grasses” is not correct as the grasses are clonal, only fairy rings are not formed by clonal growth. Clonal growth is defined as potential for the production of physically independent individuals from one genetic individual. As grasses lack main root and produce adventitious roots on their belowground stems (rhizomes) they are potentially (and also practically) clonal (there are exceptions in annual grasses that form only one tiller during their lifespan, e.g. *Zea mays* and also other annuals).

Jitka Klimesova